# MEOX2 Transcription Factor Is Involved in Survival and Adhesion of Glioma Stem-like Cells

**DOI:** 10.3390/cancers13235943

**Published:** 2021-11-25

**Authors:** Gaëlle Tachon, Konstantin Masliantsev, Pierre Rivet, Amandine Desette, Serge Milin, Elise Gueret, Michel Wager, Lucie Karayan-Tapon, Pierre-Olivier Guichet

**Affiliations:** 1Université de Poitiers, CHU Poitiers, ProDiCeT, 86000 Poitiers, France; gaelle.tachon@chu-poitiers.fr (G.T.); konstantin.masliantsev@chu-poitiers.fr (K.M.); amandine.chepied@univ-poitiers.fr (A.D.); michel.wager@chu-poitiers.fr (M.W.); 2Laboratoire de Cancérologie Biologique, CHU Poitiers, 86000 Poitiers, France; pierre.rivet@chu-poitiers.fr; 3Service d’Anatomo-Cytopathologie, CHU Poitiers, 86000 Poitiers, France; serge.milin@chu-poitiers.fr; 4Université Montpellier, CNRS, INSERM, 34094 Montpellier, France; elise.gueret@mgx.cnrs.fr; 5Montpellier GenomiX, France Génomique, 34095 Montpellier, France; 6Service de Neurochirurgie, CHU Poitiers, 86000 Poitiers, France

**Keywords:** MEOX2, glioma stem-like cells, differentiation, ERK, CDH10

## Abstract

**Simple Summary:**

Glioblastoma is the most common and lethal primary brain tumor for which no curative treatment currently exists. In our previous work, we showed that MEOX2 was associated with a poor patient prognosis but its biological involvement in tumor development remains ill defined. To this purpose, the aim of our study was to investigate the role of MEOX2 in patient-derived glioblastoma cell cultures. We unraveled the MEOX2 contribution to cell viability and growth and its potential involvement in phenotype and adhesion properties of glioblastoma cells. This work paves the way toward a better understanding of the role of MEOX2 in the pathophysiology of primary brain tumors.

**Abstract:**

The high expression of MEOX2 transcription factor is closely associated with poor overall survival in glioma. MEOX2 has recently been described as an interesting prognostic biomarker, especially for lower grade glioma. MEOX2 has never been studied in glioma stem-like cells (GSC), responsible for glioma recurrence. The aim of our study was to investigate the role of MEOX2 in GSC. Loss of function approach using siRNA was used to assess the impact of MEOX2 on GSC viability and stemness phenotype. MEOX2 was localized in the nucleus and its expression was heterogeneous between GSCs. MEOX2 expression depends on the methylation state of its promoter and is strongly associated with *IDH* mutations. MEOX2 is involved in cell proliferation and viability regulation through ERK/MAPK and PI3K/AKT pathways. MEOX2 loss of function correlated with GSC differentiation and acquisition of neuronal lineage characteristics. Besides, inhibition of MEOX2 is correlated with increased expression of CDH10 and decreased pFAK. In this study, we unraveled, for the first time, MEOX2 contribution to cell viability and proliferation through AKT/ERK pathway and its potential involvement in phenotype and adhesion properties of GSC.

## 1. Introduction

Glioma stem-like cells (GSC) is a subpopulation of cells present in the tumor with characteristics similar to the neural progenitor cells [1,2]. GSC present fundamental stem characteristics consisting of self-renewal, multipotency, and high tumorigenicity [3,4]. These cells can survive exogenous DNA damage such as radiation-induced double strand breaks or chemotherapeutic drugs and repopulate the tumor following treatment, thereby contributing to radiotherapy resistance, chemoresistance and tumor recurrence [5,6].

Growing evidence suggests that transcription factors play a major role in maintaining the stem or differentiating properties of cells [7]. One study identified 19 transcription factors, involved in neurodevelopment and selectively expressed in GSC, which maintained stemness phenotype and prevented differentiation [8]. A subset of only four of them, SOX2, OLIG2, POU3F2, and SALL2, was sufficient to completely reprogram differentiated cells into GSC. Homeobox genes, encoding transcription factors, have been described as major player of stem property regulation during embryonic development and in cancers [9]. Among homeobox genes involved in cancer, MEOX2 has recently been described as a possible actor of carcinogenesis in Wilms tumor, lung cancer, and in laryngeal cancers [10,11,12]. In non-small cell lung cancer, Ávila-Moreno et al. showed that the overexpression of MEOX2 was correlated with poor patient survival. Moreover, its overexpression was responsible for chemoresistance of A-427 and INER-37 cell lines to cisplatinium.

In gliomas, Bao et al. analyzed TCGA dataset for 117 glioblastomas (GBM) with mesenchymal subtype and identified a molecular signature of 17 transcription factors, including MEOX2, which correlated with overall survival [13]. Turcan et al. analyzed large-scale epigenomic modifications of glioblastoma and astrocyte gliomaspheres, induced by the p.R132H mutation of IDH1 [14]. The occurrence of p.R132H mutation of IDH1 is an early mechanism of gliomagenesis of lower grade gliomas and secondary glioblastoma, and Turcan et al. demonstrated that *MEOX2* expression negatively correlated with *IDH1* mutation. Our team recently demonstrated that MEOX2 was a potent prognostic factor of patient outcome in all gliomas and, more interestingly, in lower grade glioma. Moreover, it appeared to be a robust prognostic marker of survival in IDH wild-type lower grade glioma [15].

Little is known about MEOX2 contribution to carcinogenesis and this study constitutes the first attempt to characterize the functional role of MEOX2 in glioma, and specifically, in GSC.

## 2. Materials and Methods

### 2.1. GSC-Lines and Cell Culture

Glioma stem cell cultures were derived from freshly resected tumors after informed consent was obtained from each patient. This study was approved by the ethics committee of Poitiers University Hospital (DHOS/OPRC/FCnotif-tumoro-jun04: 04056), in accordance with the Declaration of Helsinki. Cells were grown as previously described in [16,17,18]. Briefly, GSC were cultured at 37 °C as proliferative non-adherent spheres in Neurobasal medium (Life Technologies, Carlsbad, CA, USA) supplemented with B27, N2, and bFGF and EGF at 20 ng/mL (Life Technologies). Culture medium was replaced twice a week and, when spheres became large, they were enzymatically dissociated with accutase (Merck-Millipore, Billerica, MO, USA). Molecular traits of the GSC are specified in Appendix A. GSC cultures were assessed for stemness, self-renewal, and differentiation in vitro and tumorigenicity was evaluated in vivo by intracranial xenografts in immunodeficient mice.

### 2.2. Patient Cohort

Formalin-fixed, paraffin-embedded tissues from 37 surgical resections or biopsies of gliomas were obtained from enrolled patients operated at the University Hospital of Poitiers. The use of human tissue was granted by the secretary of state for education and research, directorate-general for research and innovation, bioethics unit (DC-2008-565), in accordance with the Declaration of Helsinki. Median age at diagnosis was 50 years. Histopathology and molecular characterization were retrieved from the clinical report database (Appendix A).

### 2.3. Transfection

MEOX2 expression was knockdown by siRNA (MEOX2 select silencer siRNA, ThermoFisher Scientific) using lipofectamine reagent RNAiMAX (Life technologies, Carlsbad, California, CA, USA) and according to manufacturer instructions. Non-targeting siRNA (Silencer^®^ Select Negative Control siRNA, ThermoFisher Scientific, Waltham, MA, USA) served as negative controls.

### 2.4. Cell Viability Assay

Cell viability was assessed using the CellTiter 96^®^ Aqueous Non-Radioactive Cell Proliferation Assay (Promega, Madison, WI, USA). Between 5000 and 10,000 cells per well were seeded in a 96-well microplate and transfected with siRNA. Quantification of viable cells was carried out by measuring the absorbance at 492 nm using micro plate reader Infinite^®^ F50 (TECAN, Männedorf, Switzerland).

### 2.5. Pyrosequencing and Methylation

The methylation profile of 3 CpG sites located in the MEOX2 promoter, at −1409 of the translation initiation codon, was investigated as previously described by Wouters et al. [19]. Briefly, DNA was extracted using the QIAamp DNA Mini Kit (Qiagen, Les Ulis, France) according to supplier’s recommendations. The DNA was treated with sodium bisulfite, using the EZ DNA Methylation-Gold Kit (Zymo Research, Irvine, CA, USA) and PCR amplification and pyrosequencing were performed on 5 µL of bisulfitized DNA using the PyroMark Q24 Gold Reagents (Qiagen) mixed with 1 µL of MEOX2 primers: MEOX2 Forward 5′-AGGGTTTTGAAGTTGTTATTTGTTT-3′, MEOX2-Biotinyl Reverse 5′-ACATAACTATTCCTCCTACTCAT-3′, and MEOX2 Pyroseq 5′-GTTTTGAAGTTGTTATTTGTTTG-3′. Each run comprised internal quality controls: a negative control and two positive controls, one of which was methylated (Methylated Human Control, Promega) and the other unmethylated (Unmethylated Human Control DNA, Qiagen). Final pyrographs were obtained using the Pyromark Q24 software (Qiagen).

### 2.6. Subcellular Fractionation and Western Blotting

Cell fractioning was performed using Cell Fractionation Kit (Cell Signaling Technology, Danvers, MA, USA) according to supplier’s instructions. Cells were washed with 1X Phosphate-buffered saline (PBS) then lysed with cold RIPA buffer (Sigma-Aldrich) supplemented with protease and phosphatase inhibitors. Protein concentration was determined by the Bradford assay using BioRad protein assay (BioRad, Hercules, CA, USA) with absorbance measurement set up at 595 nm [20]. An equal quantity of protein samples was separated by SDS-PAGE and transferred onto nitrocellulose membrane using the rapid Trans-Blot^®^Turbo™ transfer technique (BioRad). Non-specific antigenic sites of the membrane were blocked for 1 h at room temperature with PBS 1X containing 0.1% Tween 20 (Sigma-Aldrich, Saint-Quentin Fallavier, France) and 5% skim milk. Primary antibodies were incubated overnight at 4 °C and secondary antibodies were incubated for 1 h 30 min at room temperature. Immunoblotting signal detection was carried out by a chemiluminescence method (Clarity ™ Western ECL Substrate, BioRad) by the Luminescent Image Analyzer LAS-3000 (Fujifilm, Minato, Tokyo, Japan). Signal quantification was performed using ImageJ software (https://imagej.nih.gov/ij/, accessed on 1 October 2021, La Jolla, CA, USA) and normalized with β-actin. LAMIN A/C and GAPDH were used as control to identify nuclear and cytosolic fractions, respectively. The list of antibodies used is indicated in Appendix A.

### 2.7. Immunofluorescence

Coverslips were coated with 500 µL laminin 1 µg/mL and poly-d-lysine 50 µg/mL in 1X PBS or with growth factor reduced Matrigel (Corning, New York, NY, USA), diluted halfway with supplemented NBE. Cells were then seeded on coverslips for 48 h–72 h, fixed with 4% paraformaldehyde for 15 min. Blocking and cell permeabilization were performed using 1X PBS buffer containing 0.1% Triton X-100 and 5% goat serum (or 4% Bovine Serum Albumin) for 1 h at room temperature. The primary antibodies were then incubated overnight at 4 °C, and the secondary antibodies were incubated for 1 h 30 min at room temperature in the dark. Phalloidin-Alexa fluor 546 (Invitrogen, Carlsbad, CA, USA) was added to the mix. Nuclear staining was performed using DAPI (1 μg/mL) and coverslips were mounted with Fluorescence Mounting Medium DAKO (Agilent technologies, Santa Clara, CA, USA). An AxioImager microscope equipped with an Apotome module was used for imaging (Carl Zeiss, Oberkochen, Germany).

### 2.8. Cell Cycle Analysis

Cells were dissociated with accutase and suspended dropwise in cold 70% ethanol (−20 °C) for fixation. The cells were then washed with cold 1X PBS and resuspended in labeling solution containing 2.5 μg/mL of propidium iodide (PI) and 0.5 mg/mL of RNase A (Merck Millipore, Saint-Quantin-en-Yvelyne, France). Apoptosis analysis was performed with the Annexin-V-FLUOS Staining Kit (Sigma-Aldrich) following the supplier’s recommendations. Briefly, the cells were accutased, washed once with 1X PBS, then suspended in 400 µL of 1X PBS + 100 µL of incubation medium with 2 µL of annexin [1 mg/mL] and 2 µL of PI [1 mg/mL]. Cell cycle analysis and apoptosis were carried out by flow cytometry using the fluorescence-activated cell sorting (FACS) Canto II (BD Biosciences, Franklin Lakes, NJ, USA). A total of 10,000 events were analyzed for each sample using FlowJo^®^ software.

### 2.9. Transcriptome Analysis

The RNAs were extracted using the RNeasy Mini Kit (Qiagen) following the manufacturer’s protocol and qualified using the Agilent 2100 Bioanalyzer platform (Agilent Technologies). Qualified RNAs were sent to the MGX platform in Montpellier for RNASeq analysis. Briefly, libraries were produced using a TruSeq Stranded mRNA Sample Preparation Kit (Illumina, San Diego, CA, USA) and paired-end sequenced 150nt on NovaSeq sequencer (Illumina). The RNAseq data (fastq files) have been deposited in the Sequence Read Archive database under the accession number PRJNA781814. Raw data were collected and processed using the bioinformatic strategy of genome-guided alignment [21]. The mapping tool used was Hisat2, the counting tool used was FeatureCounts and the tools used for the determination of differential expression were DESeq2.

### 2.10. Statistical Analysis

Statistics of qualitative and quantitative data were performed using GraphPad Prism 6 software (San Diego, CA, USA). All experiments were performed at least 3 times and histograms represent the mean ± SEM. Statistical significance was evaluated by Kruskal–Wallis, Mann–Whitney, Fisher exact test (* *p*  <  0.05; ** *p* < 0.01; *** *p* < 0.001).

## 3. Results

### 3.1. Nuclear and Heterogeneous MEOX2 Expression among GSC

Among 16 GSC lines previously analyzed by transcriptomic approach using Agilent SurePrint G3 Human GE chip, we selected three GSC lines expressing MEOX2, GSC-2, GSC-6, and GSC-10, and 3 GSC lines negative for MEOX2 expression, GSC-11, GSC-7, and GSC-5 [18,22]. Presence or absence of MEOX2 expression was confirmed by WB approach (Figure 1a).

MEOX2 nuclear localization was confirmed by immunofluorescence and subsequently verified by cell fractionation assay (Figure 1b,c). These results suggested that the expression of MEOX2 was nuclear and heterogeneous in GSC.

Back at the tumor tissue from which GSC2, 6 and 10 were derived, IHC analysis confirmed the strong nuclear expression of MEOX2 (Figure 1d). Subsequently, expression of MEOX2 in GSC seems to be consistent with strong expression in the tumor.

### 3.2. MEOX2 Expression Is Partly Dependent on the Methylation Status of Its Promoter

We had previously demonstrated that the expression of MEOX2 in the tumor tissue correlates with its methylation profile and with wild-type *IDH1/2* [15]. Consequently, it is interesting to compare the methylation status of *MEOX2* promoter in IDH1-mutated tumors with IDH1 wild-type tumors (Figure 2a). With one exception, all IDH1-mutated gliomas were methylated, while none were methylated in the other group (Appendix A). Methylation of *MEOX2* promoter correlated with *IDH1* mutation status and was independent of tumor grade (WHO, 2016 classification) (Figure 2b, *p* < 0.001). These results provided confirmation that in gliomas, the expression of *MEOX2* is dependent on the *IDH1* mutation status.

MEOX2 expression was heterogeneous among GSC which are all wild-type *IDH1*, suggesting that its expression may depend on multiple factors. We have previously shown that in glioma, there was an inverse correlation between *MEOX2* expression and its promoter methylation status [15]. The methylation status of 3 CpG sites in the GSC-2, -6, and -10 lines were tested by pyrosequencing and compared to MEOX2 non-expressing GSC-11, GSC-7, and GSC-5 lines. The GSC-2 and GSC-10 lines were not methylated, and the average methylation of GSC-6 line was 60%, whereas the average methylation of MEOX2 non-expressing cell lines was 34%, 69%, and 32%, respectively (Figure 2c). Then, absence of MEOX2 expression in GSC-11, GSC-7, and GSC-5 could partially be explained by epigenetic mechanism.

### 3.3. MEOX2 Is Involved in Cell Survival through ERK/MAPK and PI3K/AKT Pathways

The involvement of MEOX2 in GSC was studied by loss-of-function approach on GSC-2, -6, and -10 using two different siRNA against MEOX2 and one non-silencing (NS) as control (Figure 3a). MTS assay was carried out after inhibition of MEOX2 expression for 5 days, showing a significant increase of cell viability (Figure 3b, *p* < 0.05). MEOX2 is known to modulate the ERK/MAPK pathway in the A549 cell line derived from lung carcinoma, which may suggest that MEOX2 affects cell viability may be driven by the ERK/MAPK pathway [23]. To test this hypothesis, activation of the ERK/MAPK pathway was investigated by WB in GSC-2 and GSC-6 with or without MEOX2 knockdown. MEOX2 inhibition was associated with ERK/MAPK pathway activation, as demonstrated by phosphorylation of the T202/Y204 sites of ERK1/2 (Figure 3c, *p* < 0.05).

As MEOX2 loss of expression increases viability, it may also reduce the percentage of cells in necrosis or apoptosis. Analysis of apoptosis by FACS after MEOX2 inhibition for 48 h or 5 days did not show significant decrease in the percentage of PI positive cells in early/late apoptosis/necrosis phase (Figure 4a). However, a decrease of cleaved PARP was found in GSC-6 (Figure 4b; *p* < 0.05). In various cell models, MEOX2 has been reported to exhibit pro-apoptotic properties through PI3K/AKT pathway inhibition [12,24]. Inhibition of MEOX2 for 48 h led to an increase of pT308-AKT, which reflected the activation of PI3K/AKT pathway (Figure 4c; *p* < 0.05). Taken together, these results show that MEOX2 has anti-proliferative effect in GSC by the modulation of ERK/MAPK and PI3K/AKT pathways.

### 3.4. The Loss of MEOX2 Expression Correlates with the Differentiation of GSC and with Acquisition of Neuronal Lineage Features

In quiescent vascular smooth muscle cells, the addition of serum decreased the expression of MEOX2 [25]. As previously reported, addition of serum in GSC cultures causes their differentiation, as corroborated by our results showing the decrease of stemness marker OLIG2 and the increase of differentiation markers GFAP and B-tubuline III, in GSC-2 and GSC-6 cultured with 5% fetal bovine serum (Appendix A) [26,27,28,29]. Interestingly, under these conditions, the expression of MEOX2 decreased (Figure 5a; *p* < 0.05) without changes in methylation profile of the *MEOX2* promoter (Figure 5b). MEOX2 could then have an effect on differentiation process of GSC. MEOX2 inhibition for 6 days was responsible for the decrease of oligodendrocyte marker OLIG2 and the increase of neuronal differentiation markers MAP2 and DCX in GSC lines (Figure 5c; *p* < 0.05). These results suggest that the inhibition of MEOX2 would preferentially lead to GSC neuronal differentiation.

### 3.5. Inhibition of MEOX2 Regulates a Restricted Gene Network including Cadherin CDH10

To further study regulatory networks and biological processes affected by MEOX2 loss of function, a differential transcriptome analysis comparing GSC-2, -6, and -10 transfected with siRNA1- and siRNA2-MEOX2, versus GSC transfected with non-silencing siRNA (nsRNA) was performed (Figure 6a).

The expression of four genes was systematically modified in the same direction by the two siRNAs in the three cell lines (Figure 6b,c). Among these four genes, one was of particular interest—CDH10 encoding for cadherin 10, also called T2-cadherin. Indeed, in adulthood, CDH10 is almost exclusively expressed by the brain and is thought to play a role in the migration and adhesion of synapses and axons [30]. While its role in cancers is unknown, compared with other tumors, it is mostly expressed in gliomas [15]. In GSC, CDH10 expression increased after 48 h of MEOX2 knockdown (Figure 7a; *p* < 0.05).

In stem cells, cadherins play a role in cell–cell adhesion, particularly in the stromal niche, where loss of cadherin expression leads to detachment from their supporting cap cells [31]. To assess whether MEOX2 has an impact on adhesion properties through the cadherin pathway, we cultured GSC on laminin/D-lysine or matrigel coating. Under adhesion condition, CDH10 upregulation was accompanied by change in the CDH10 protein localization. In MEOX2-inhibited GSC, CDH10 was partially located at the cytoplasmic membrane, whereas in basal GSC, CDH10 was localized in the nucleus (Figure 7b). This change in location may reflect the functional activation of CDH10. In view of these results and data from the literature, while downregulating CDH10 expression, MEOX2 could influence GSC’s adhesion to a matrix [32].

In MEOX2 knockdown GSC-2 and GSC-6 cultured on laminin/D-lysine coating for 24 h, the WB analysis showed a decrease of pY397-FAK/FAK ratio, a specific marker of cell adhesion, mediated by integrins (Figure 7c; *p* < 0.05) [33].

Taken together, these results suggest that MEOX2 may regulate a restricted gene network comprising CDH10 and that, in GSC, MEOX2 may play a role in the cell–cell matrix adhesion process.

## 4. Discussion

GSC are cause of resistance to chemo/radiotherapy and are responsible for glioma recurrence. Increasing evidence suggests that GSC fate decisions can be overridden by the artificial expression of small sets of transcription factors [7,8]. We previously demonstrated that the transcription factor MEOX2 was a robust prognostic factor, which correlated with overall survival and progression-free survival in patients with glioma. A couple of months later, the prognosis interest of MEOX2 was confirmed by an independent team [34]. Nevertheless, MEOX2 influence on GSC fate remained unknown. To our knowledge, this study is the first attempt to characterize the functional role of MEOX2 in glioma, particularly in GSC.

Not every GSC shares MEOX2 expression, approximately half carry this trait. Stem cells are very heterogeneous and difference of transcription factor expression between GSC has already been observed [35,36]. Indeed, Gallo et al. compared gene expression in a set of 27 primary glioblastoma neural stem cell cultures and 12 fetal neural stem cell cultures. They concluded that more than half of glioblastoma neural stem cells were characterized by high expression of HOX genes, whereas the other half was not. In our study, MEOX2 expression in GSC was consistent with its expression in the original tumor. This result would tend to show that the expression of MEOX2 in GSC reflects the expression of this gene in the rest of the tumor tissue. A transcription factor, such as MEOX2, must be localized at nuclear level to be active. When expressed, MEOX2 was well localized in the nucleus in GSC and in tumor tissue. Based on the loss-of-function approach, MEOX2 in GSC appeared to inhibit proliferation through ERK/MAPK pathway downregulation. Similar results have been reported in another cell line, the A549 derived from lung carcinoma [23]. In this cell line, the authors showed that MEOX2 inhibition was responsible for the activation of CREB, an effector of the MAPK signaling cascade. Another study on adventitial fibroblasts found a similar correlation [37]. According to our results, MEOX2 may also enhance apoptosis through PI3K/AKT pathway downregulation. This observation was consistent with the literature. Indeed, Tian et al. reported that overexpression of MEOX2 promoted apoptosis through inhibiting the PI3K/AKT pathway in laryngeal cancer cells [12]. Likewise, in perivascular adipocytes, MEOX2 was responsible for early and late apoptosis enhancement through downregulation of PI3K/AKT1 and ERK1/2 signaling pathways [24]. Finally, Yin et al. drew similar conclusions using smooth chicken muscle cell models [38].

In quiescent vascular smooth muscle cells, the addition of serum or other mitogens in culture cell decreased *MEOX2* expression [25]. In this study, our results suggest that (i) during GSC differentiation, the expression of *MEOX2* would decrease by mechanism independent from the methylation of its promoter; (ii) the targeted inhibition of MEOX2 would lead to preferential differentiation of GSC into neuronal lineage; (iii) MEOX2 could play a role in the equilibrium between stemness/neuronal differentiated phenotype. Indeed, in GSC, induction of differentiation by adding serum to the culture medium resulted in decreasing MEOX2 expression in all three cell lines. A similar result was observed in the study by Berezovsky et al., in which the knockdown of SOX2, a major actor of the stemness phenotype, resulted in loss of expression of MEOX2 [39]. Finally, another study reported higher expression of MEOX2 in GBM and tumor-derived GSC, compared with conventional cell lines [40]. In GSC, MEOX2 appeared to be an active player in the balance between stemness and neuronal phenotypes. Similarly, in patient-derived GBM lines, downregulation of HOXA9 also resulted in enrichment of genes involved in neuronal differentiation [41]. The fact that specific transcription factors may be responsible for stemness phenotype maintenance is not new. During embryogenesis, neuroepithelial stem cells were able to downregulate their epithelial features by assistance from homeobox genes such as Forkhead Box G1 (FOXG1), LIM Homeobox 2 (LHX2), Paired Box 6 (PAX6), and Empty Spiracles Homeobox 2 (EMX2) to generate radial glia [42]. Another study identified 19 transcription factors, involved in neurodevelopment, which were selectively expressed in GSC to maintain their stem phenotype and prevent their differentiation [8]. Finally, Gallo et al. identified a HOX signature that was specific to GSC in comparison with 2 human fetal neural stem cell lines and 6 resections of non-neoplastic brain tissue [36].

No major biological function appeared to be impaired by loss of MEOX2 expression in GSC, only the expression of a few independent genes was modified. In a similar study, Pojo et al. carried out a transcriptomic analysis comparing commercial cell lines—U87 and U251—an immortalized astrocyte line and a GSC line, respectively, and showed that only a small subset of probes was consistently altered by HOXA9 knockdown [41]. MEOX2 knockdown was responsible for increasing CDH10 expression, which encodes for cadherin-10 protein (also called T2-cadherin). In cancers, cadherins rather tend to play a role of tumor suppressor genes and their expression is often lost during carcinogenesis [43]; however, during the epithelio–mesenchymal transition, a cadherin “switch” mechanism has been reported with E-cadherin expression decreasing while N-cadherin expression is increasing [44]. However, E-cadherin is little expressed in gliomas [45,46]. On the contrary, CDH10 is largely expressed in glioma and its expression is higher in quiescent mouse neural stem cells compared with non-quiescent EGFR^+^ neural stem cells [47]. In GSC cultured on matrix, MEOX2 knockdown was associated with decreasing phosphorylation of FAK, which implied a restraint in focal adhesion mediated by integrin pathway. Pojo et al. also showed that downregulation of a homeobox gene HOXA9 in the U251, and in GSC lines, induced an increase of cell adhesion markers [41]. Similarly to MEOX2, at tumor scale, HOXA13 upregulation was associated with enhancement of focal adhesion mediated by integrins and reduction of MAPK pathway activation [48]. Our results suggest that in GSC, MEOX2, through CDH10 downregulation, impacts mechanisms of adhesion by promoting focal adhesion, contrary to cell–cell adhesion.

## 5. Conclusions

Taken together, this study showed that MEOX2 is mainly involved in cell proliferation and survival through the regulation of ERK/AKT pathway. Moreover, it could be associated with GSC phenotype maintenance by repressing neuronal lineage differentiation. Finally, MEOX2 could regulate GSC adhesion properties, notably by modulating CDH10 expression and FAK activation.

## Figures and Tables

**Figure 1 cancers-13-05943-f001:**
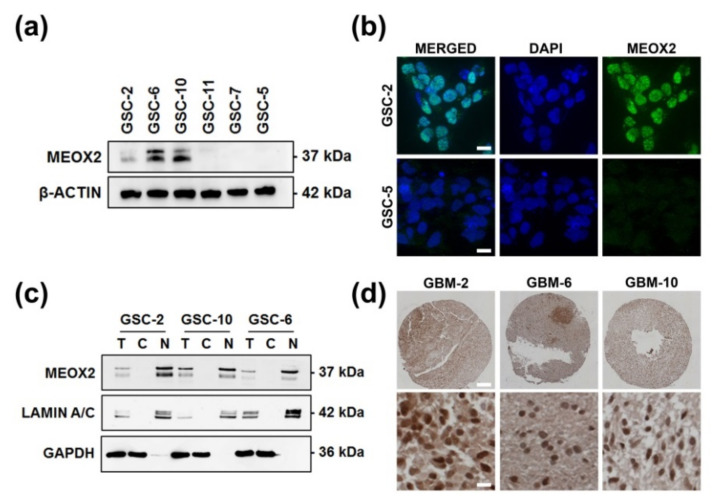
MEOX2 expression and localization among GSC. (**a**) Basal expression of MEOX2 in six GSC; (**b**) Representative immunofluorescence images of MEOX2 (green) in GSC-2 (dapi = blue C, Scale bar: 10 µm); (**c**) Cell fractioning analysis of MEOX2 location in GSC-2, GSC-6, and GSC-10 lines (T = total, C = cytoplasmic, N = nuclear) with LAMIN A/C and GAPDH as controls corresponding to nuclear and cytoplasmic fraction respectively; (**d**) Immunohistochemical analysis of MEOX2 expression in tumors from which GSC originated (Scale bar =100 µm for high panel and =10 µm low panel).

**Figure 2 cancers-13-05943-f002:**
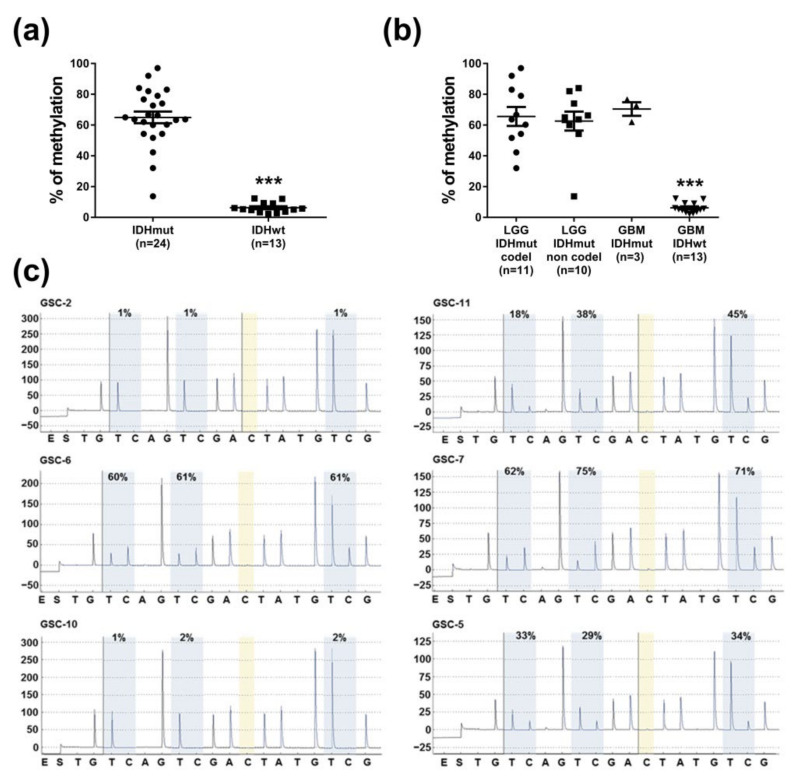
Methylation status of *MEOX2* promoter. (**a**) Average percentage of methylation according to *IDH1* mutational status; (**b**) Average percentage of methylation for each molecular subtype of glioma (*n* = 37), according to the 2016 WHO classification; *** *p* < 0.001, Kruskal–Wallis test; (**c**) Promoter methylation profile of three CpG sites in MEOX2-expressing (2, 6, 10) and non-expressing GSC (11, 7, 5).

**Figure 3 cancers-13-05943-f003:**
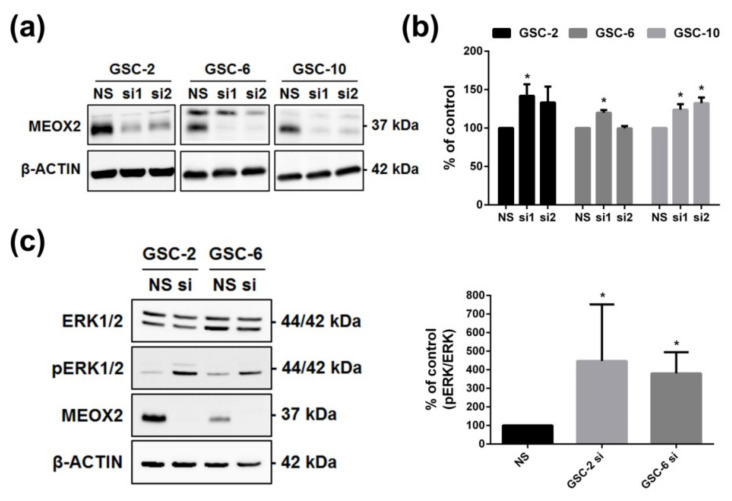
Proliferative effect of MEOX2 through the ERK/MAPK pathway. (**a**) Western blot validation of MEOX2 siRNA inhibition; (**b**) Analysis of GSC-2, -6, and -10 viability using two siRNA, 5 days after inhibition of MEOX2 expression (*n* = 4, *p* < 0.05, Mann–Whitney test); (**c**) Representative WB and histograms representing the variation of pERK/ERK after inhibition of MEOX2 (siRNA1) for 48 h in GSC-2 and -6 (*n* = 4, * *p* < 0.05, Mann–Whitney test).

**Figure 4 cancers-13-05943-f004:**
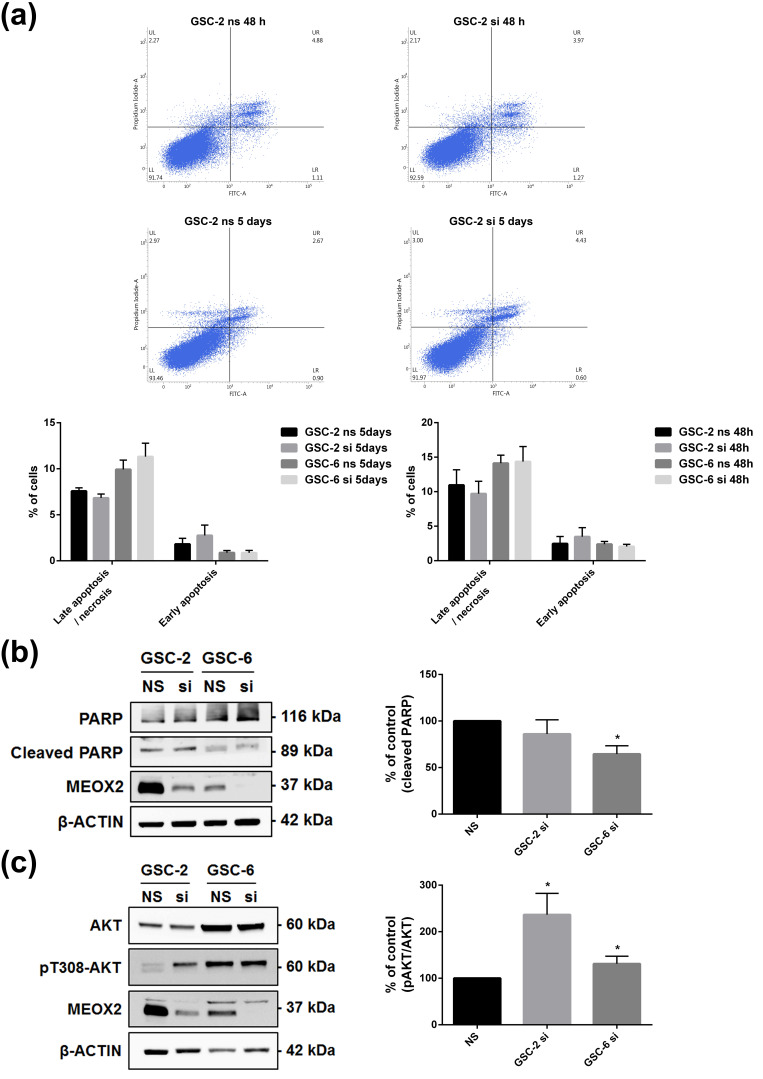
Cell survival regulation by MEOX2 through PI3K/AKT pathway. (**a**) Representative plots of FACS analysis of Annexin V/PI after inhibition of MEOX2 (siRNA1) for 48 h or 5 days in GSC-2; (**b**) Representative WB and histograms of cleaved-PARP decrease after inhibition of MEOX2 (siRNA1) for 48 h (*n* = 4, * *p* < 0.05, Mann–Whitney test); (**c**) Representative WB and histograms of the activation of AKT pathway after inhibition of MEOX2 (siRNA1) for 48 h in GSC-2 and GSC-6 (*n* = 4, * *p* < 0.05, Mann–Whitney test).

**Figure 5 cancers-13-05943-f005:**
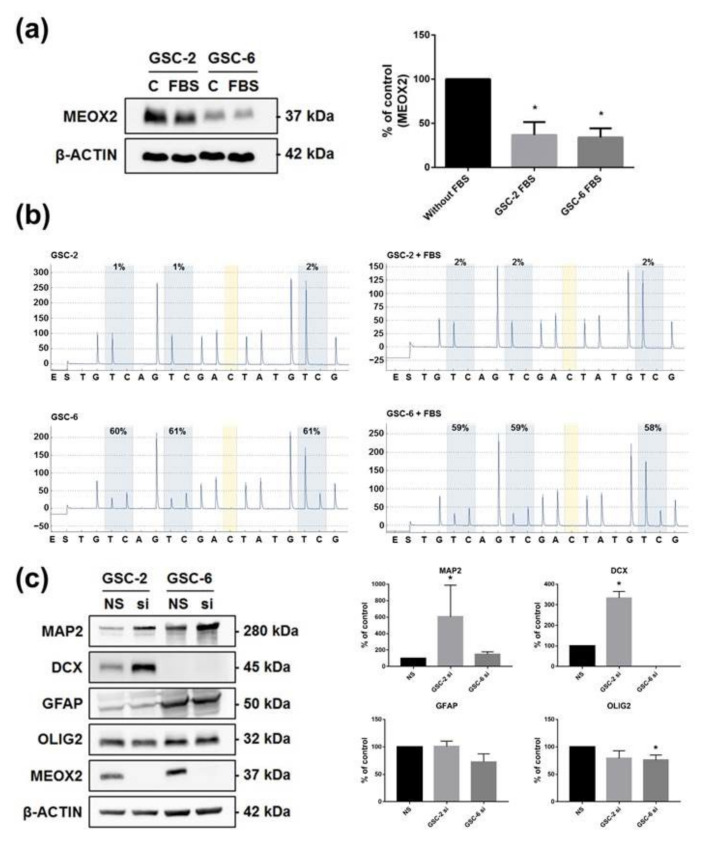
Differentiation of GSC and acquisition of neuronal lineage features after MEOX2 inhibition. (**a**) Representative WB and histograms of MEOX2 expression after 6 days of culture in 5% Fetal Bovine Serum (FBS) (*n* = 4, * *p* < 0.05, Mann–Whitney test); (**b**) methylation profile of GSC-2, -6 lines after 6 days of culture in 5% fetal bovine serum (FBS); (**c**) representative WB and histograms for MAP2, DCX, GFAP, and OLIG2 expression after inhibition of MEOX2 (siRNA1) for 6 days in GSC-2 and GSC-6 (*n* = 4, * *p* < 0.05, Mann–Whitney test).

**Figure 6 cancers-13-05943-f006:**
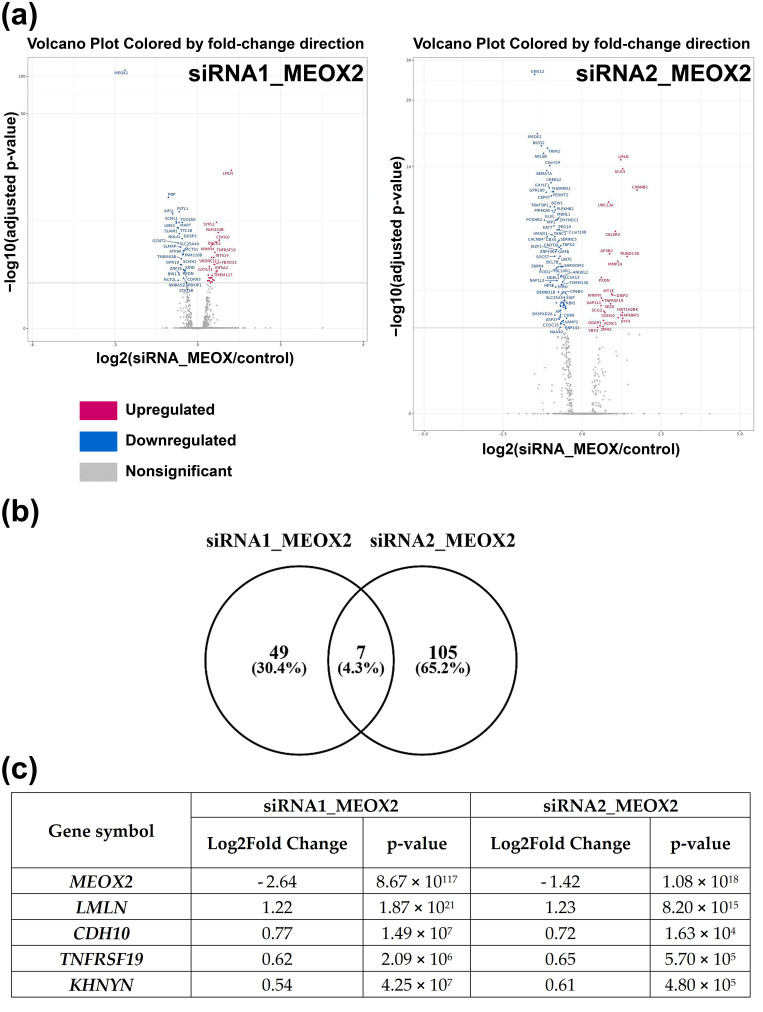
RNAseq analysis of MEOX2-regulated genes in GSC-2, -6, and -10. (**a**) Volcano plots of significant differentially expressed genes after MEOX2 inhibition by two different siRNA; (**b**) Venn diagram of common differentially expressed genes between two siRNA; (**c**) List of four common genes affected in same way by MEOX2 inhibition.

**Figure 7 cancers-13-05943-f007:**
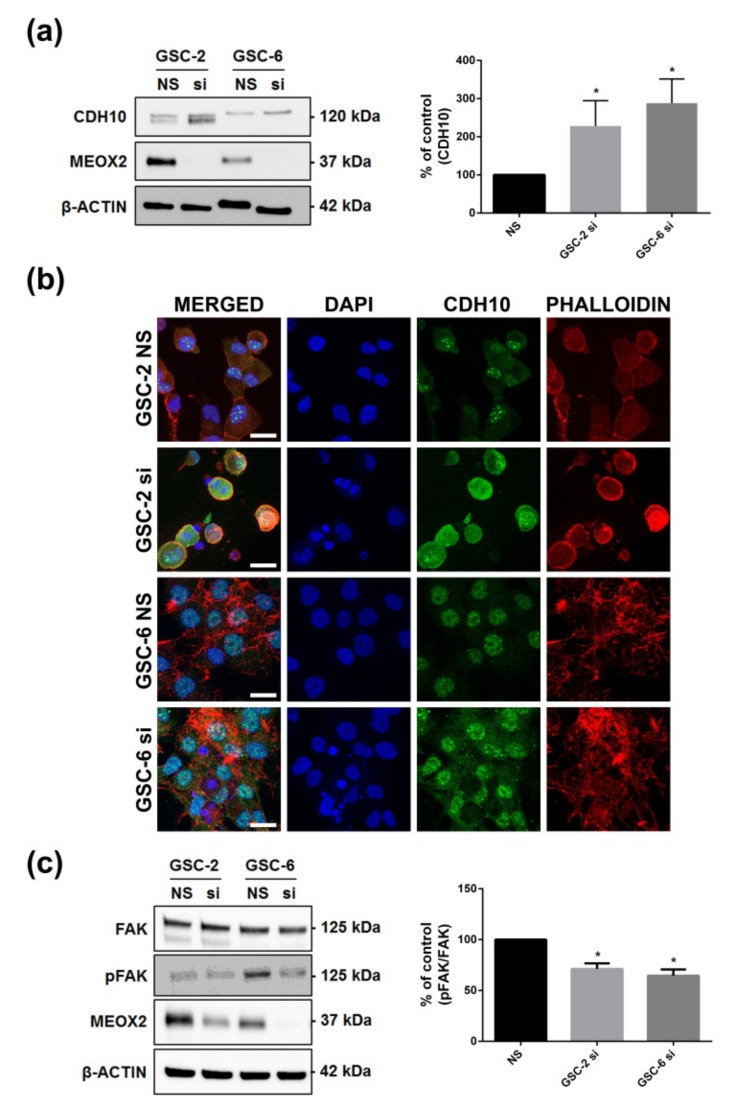
MEOX2 inhibition modifies CDH10 expression and FAK activation. (**a**) Representative WB and histograms of CDH10 after 48 h of MEOX2 inhibition by siRNA1 in GSC-2 and GSC-6 (*n* = 4, *p* < 0.05, Mann–Whitney test); (**b**) Representative immunofluorescence images of the expression of CDH10 (green) after 48 h of MEOX2 knockdown in GSC-2 and GSC-6 (dapi = blue; phalloidin = red; scale bar = 10 µm); (**c**) Representative WB and histograms of p-FAK/FAK expression after 48 h MEOX2 knockdown (siRNA1) in the GSC-2 and GSC-6 lines cultured under adhesive condition (*n* = 4, * *p* < 0.05, Mann–Whitney test).

## Data Availability

All the data and material are available on reasonable request from the corresponding author.

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
