# Peer review of "MEOX2 Transcription Factor Is Involved in Survival and Adhesion of Glioma Stem-like Cells"

_cancers, 2021, doi:10.3390/cancers13235943_

Round 1

Reviewer 1 Report

Comments to the authors

The manuscript “The role of the transcription factor MEOX2 in glioblastoma stem cells” by Gaëlle Tachon et. al., addresses the role of the transcription factor MEOX2 in some characteristics of glioma stem cells such as proliferation, differentiation, and the expression and activation of proteins related to invasive capacity. The results of this study are of great novelty and provide valuable information in the context of glioma stem cells. While this manuscript covers an interesting topic, there are some major and minor drawbacks that must be improved.

Major revisions

Abstract and other sections:

In the Abstract, Results, and Discussion sections, the authors mention that MEOX2 expression depends on methylation status. This is true if they added that there is a dependence on IDH1 mutations.

In the Abstract, Results, and Discussion sections, authors cannot assume that MEOX2 exhibits pro-apoptotic effects since the results of these experiments were not statistically significant.

At the end of the Abstract, make a conclusion about the main results of your work.

Results:

Which is the percentage of GSC in the tumors?

Why MEOX2 presents 2 bands?

Why the intensity of MEOX2 and lamin is lower in the total cells than in the nuclear fraction?

It is mentioned that MEOX2 inhibition decreases astrocytic markers but changes in GFAP were not statistically significant.

The authors say: “Analysis of apoptosis by FACS after MEOX2 inhibition for 48 h or 5 days tended…” The authors should demonstrate that after 5 days the siRNA efficiency is maintained.

The authors say: “...to show a decrease in the percentage of PI-positive cells, and therefore of cells in late apoptosis/necrosis phase (Figure 4a).” However, in the graphs that correspond to this figure, the statistical difference is not reported, and there does not seem to be a difference between the compared groups.

In Figure 4b the decrease in the cleaved form of PARP is not clear as the total decreases proportionally.

Could the authors explain why is there no standard deviation in the control groups in some experiments (figures 3b, 4b and c, 5a and c, 6a and c)?

Discussion

The conclusions at the end of the Discussion section are very vague; the authors should emphasize their most important results and the repercussions they may have in the future in the context of glioma stem cells.

Minor revisions

Title: The title of the manuscript is unspecific; the authors should include some keywords and specify the role of MEOX2 in glioma stem cells.

Abstract and other sections:

In several paragraphs of the manuscript the word “impact” is confuse, please use, for example, increase or decrease.

In the Abstract, change glioblastoma stem cells for glioma stem cells. In fact, authors define GSC as glioma stem cells.

The authors say: “The expression of the transcription factor MEOX2 is closely associated with overall survival in glioma.” The authors should clarify whether this close association with survival is positive or negative.

Materials and Methods:

Transfection: The authors should specify the time the cells remained in contact with the siRNA.

Change bisulfated with sodium bisulfite for treated with sodium bisulfite.

Statistical analysis: The authors must specify which type of statistic corresponds to each figure.

Results:

The authors say: “The involvement of MEOX2 in GSC was studied by loss-of-function approach on GSC-2, 6 and 10 using siRNA (Figure S1).” It is suggested that at least one of the siRNAs efficiency figures appear in the main material.

There are many elements of discussion in the Results section and in some cases, the information is repeated, a re-structuring between these two sections could make the work clearer.

In Figure 6a the actin and MEOX2 blots are interchanged.

Reviewer 2 Report

The reviewed manuscript entitled ‘The role of the transcription factor MEOX2 in glioblastoma stem cells’ aims to characterize the functional role and regulatory networks and processes affected by MEOX2 in glioma, and in particular in GSC.

The authors have demonstrated in their previous study that MEOX2 is a potent prognostic factor of patient outcome in gliomas, especially in lower grade glioma. Moreover, they also showed that MEOX2 can be regarded as a robust prognostic marker of survival in IDH wild-type lower grade glioma. The reviewed study adds further knowledge into the understanding of the role of MEOX2 in glioma, especially in GSC. The results of this study demonstrate that MEOX2 impaired viability of GSC is through downregulation of ERK/MAPK pathway and increased apoptosis is a result of downregulation of PI3K/AKT pathway. In addition they show that MEOX2 seems to play a role in the conservation of the neuronal phenotype and in the regulation of cadherin 10 (CDH10) expression. They also show that MEOX2 expression depends on DNA methylation.

The study received the approval of the appropriate ethical commities. It is generally well designed and presented in a clear, comprehensive manner. The fact that the analysis were conducted using GSC lines, instead of commercially available glioblastoma cell lines is of great importance. The chosen methods are appropriate and the results are presented in a clear, organized manner. The quality of figures is generally good. The paper shows novel finding and therefore deserves publication. However some issues have to be resolved first.

Minor comments:

Why did the authors choose the particular time points, 48 h and 5 days (e.g. apoptosis analysis), 6 days (differentiation of GSC)?

Tumor material for GSC-7 lines went missing. –what exactly do you mean by ‘missing’?

‘Besides, inhibition of MEOX2 correlated with increase expression of one actor of the cadherin pathway, CDH10, and with decreased expression of one actor of the integrin pathway, pFAK.’ – to informal; I suggest identifying the role of CDH10 and pFAK more precisely, or just removing the informal ‘one actor’.

multipotency feathers??

(PBS --> brackets missing

Table S1 --> ‘Chimiotherapy (Temozolomid)’ --> Chemiotherapy (Temozolomid)

MEOX2 promoter, IDH1 mutation --> make sure gene names are written with italics, while protein names straight. Correct in the entire manuscript.

Fig. 2A – Please provide a better quality figure (the nucleotides on the x axis and numbers on y axis are hardly visible. Maybe present the Figure in the whole page width.

more alive cells --> increased viability or synonym

48 hours, 48h --> please unify

5days, 48h in text and figures --> add space between the number and ‘days’, ‘h’ etc.

neuronal differentiation, page 9 --> finish the sentence and then place the Figure and caption

non- silincing --> non-silencing

in particularly in GSC --> in particular in GSC

Reviewer 3 Report

In the manuscript by Tachon et al., the authors analyzed to role of the transcription factor MEOX2 in GSCs.

Major Points:

  • the author should preferably use the term glioma stem-like cells rather than glioma stem cells, since this is an important distinction and helps to alleviate to controversy around the term cancer stem cells. There is now ample evidence that some glioma cells can obtain a phenotype that is similar to the one seen in actual (i.e. healthy) stem cells.
  • Mat/Met: Why was N2 only used at 0.5%? Do the authors mean volume per volume (i.e. 500 µl N2 per 100?). Usually B27 and N2 are supplied as 50x and 100x concentrations. maybe the authors should check how they prepared their medium. Similarly FGF und EGF should be stated as the concentration and not the volume-fraction...
  • Page 5: Why is there a link to a dataset? Please cite the respective paper and move experimental details to the Mat/Met section.
  • Fig.1B doesn't really provide much information in particular considering the western data shown in Fig. 1c. Please either provide proof of negative stain in non-expressers (preferably including quantitative assessment of expression) or consider removing this data.
  • Fig. 1D: Pleae provide additional higher magnification images from the biopsies, because not much can be seen.
  • Chapter 3.2: The conclusion is not correct. The results do not confirm that MEOX2 expression is dependent on the methylation status. One out of 3 expressers has a very high methylation. Besides the analyses of only three samples via pyrosequencing is insufficient to draw any meaningful conclusion. In vivo it looks differently and I am wondering 1) Why this data is not presented in the main text? 2) How valuable the cell-based analyses is, if it doesn't reflect the patient data? Please revise and comment on this
  • Chapter 3.3 and the following: GSCs are particularly difficult to transfect. Thus please provide the transfection rate e.g. by using fluorescently labeled siRNA. Generally it would be preferable to employ a stable system for example using lentiviral transduction of shRNA. Is there any biological reason why the authors chose to use siRNA?
  • The change in apoptosis is not convincing. If the authors propose reduced sensitivity towards apoptosis induction the authors should perform treatment experiments using known inducers of apoptosis or cell death in general as well radiation treatment. Similarly, PARP cleavage usually requires an external stimulus and these experiments should also be repeated under treatment.
  • Fig. 5C: The quantification data should be shown in a separate graph per protein, each with its own axis. This increases the visibility, in particular the extent of inhibition. Why were no positive controls, such as serum addition (as in Fig. 5 A and B) employed? These experiments need to be repeated with the proper controls.
  • The presentation of the transcriptomic data is sub-standard. More emphasis should taken to present these data. First the effect of KD per cell line should be shown, analyzed and discussed and only then does it make sense to look at the cross cell line comparison. Additionally, is it possible that similar processes might be regulated by siMEOX2 which requires a thorough analyses of these datasets.
  • Fig. 6B: Please provide more quantititave data next to the IF pictures.

Minor points:

  • The supplements contain some non-english word (e.g. chimiotherapy). Please carefully double-check.
  •  

Round 2

Reviewer 1 Report

Comments to the authors:

The authors of the manuscript entitled “MEOX2 transcription factor is involved in survival and adhesion of glioma stem-like cells.” corrected and improved almost all deficiencies in this paper. However, there is one aspect that must be reviewed again:

In Figure 4b the decrease in the cleaved form of PARP is not clear since the total PARP decreases proportionally.

Reviewer 3 Report

all major points have been resolved.

Minor points:

  • Fig4. The heading should be re-phrased (... "by" MEOX2 ... ??)
  • Fig. 5c: Different target proteins should be presented as separate sub-figures
  • Transcriptomic data (Fig. S2) should be moved into the main text and uploaded to a public server (GEO?) prior publication.

Thank you very much!
